# Mantle data imply a decline of oxidizable volcanic gases could have triggered the Great Oxidation

Shintaro Kadoya [1⊠], David C. Catling [1], Robert W. Nicklas[2], Igor S. Puchtel[3] & Ariel D. Anbar[4]

Aerobic lifeforms, including humans, thrive because of abundant atmospheric $O_2$, but for much of Earth history $O_2$ levels were low. Even after evidence for oxygenic photosynthesis appeared, the atmosphere remained anoxic for hundreds of millions of years until the ~2.4 Ga Great Oxidation Event. The delay of atmospheric oxygenation and its timing remain poorly understood. Two recent studies reveal that the mantle gradually oxidized from the Archean onwards, leading to speculation that such oxidation enabled atmospheric oxygenation. But whether this mechanism works has not been quantitatively examined. Here, we show that these data imply that reducing Archean volcanic gases could have prevented atmospheric $O_2$ from accumulating until ~2.5 Ga with ≥95% probability. For two decades, mantle oxidation has been dismissed as a key driver of the evolution of $O_2$ and aerobic life. Our findings warrant a reconsideration for Earth and Earth-like exoplanets.

[1] Department of Earth and Space Sciences/Cross-Campus Astrobiology Program, University of Washington, Box 351310, Seattle, WA 98195-1310, USA. [2] Geoscience Research Division, Scripps Institution of Oceanography, La Jolla, CA 92093, USA. [3] Department of Geology, University of Maryland, College Park, MD 20742, USA. [4] School of Earth and Space Exploration and School of Molecular Sciences, Arizona State University, Tempe, AZ 85287, USA. ⊠email: shintaro.kadoya@gmail.com

The geological record of mass-independent sulfur isotope fractionation shows that $O_2$ first inundated the atmosphere during the Great Oxidation Event (GOE) between 2.4 and 2.1 Ga[1]. However, redox-sensitive iron and molybdenum isotope data suggest the presence of $O_2$ in the 3.2–3.0-Ga marine photic zone[2,3], which implies that $O_2$-producing cyanobacteria existed long before the GOE. Indeed, models show that, under a globally anoxic atmosphere, cyanobacteria-derived $O_2$ produces photic zone oxygen oases[4,5]. Geological data also indicate the presence of methanotrophy and oxidative nitrogen cycling in Neoarchean oceans and lakes, which suggest that $O_2$ was oxidizing sulfides and ammonium to make sulfate and nitrate, respectively[6–8]. Such inferences are consistent with phylogenetic interpretations that oxygenic photosynthesis evolved by the mid-Archean [e.g., ref. [9]]. In addition, paleoredox proxies suggest $O_2$ transients at 2.5 Ga[6,7,10–16], and point to the possibility of pervasive oxygenation of the oceans over continental margins[17,18].

The reasons for the delay of oxygenation of the atmosphere for hundreds of millions of years after the advent of oxygenic photosynthesis remain unknown. Numerous hypotheses rely on the notion that the atmospheric $O_2$ level is determined by a kinetic balance between $O_2$ supply and consumption[19–23]. If the rapid and efficient $O_2$ sinks are larger than the $O_2$ supply, there are excess reductants, and the atmosphere remains anoxic even in the presence of global oxygenic photosynthesis. Rapid sinks include net reactions of $O_2$ with reducing gases emanating from the solid Earth, e.g., $H_2$, CO, $H_2S$, $SO_2$, and $CH_4$.

One possibility is that an increase in $O_2$ fluxes overwhelmed the efficient $O_2$ sinks, causing the GOE. However, the available data are not particularly persuasive. Photosynthetic $O_2$ production is accompanied by organic matter ($CH_2O$) in the net reaction $CO_2 + H_2O = CH_2O + O_2$. Because respiration or oxidative decay reverse this reaction (presently in ~50 years), the long-term net flux of $O_2$ occurs when organic matter is segregated from $O_2$ and buried.

Long-term changes in the burial rate of organic matter—and hence in the net supply of $O_2$—can be inferred from C isotopes in sedimentary rocks. Organic matter concentrates $^{12}C$ relative to $^{13}C$, leaving marine inorganic carbonate $^{12}C$ depleted; as such, carbon isotopes in marine carbonates and kerogens track $f_{org}$, the fraction of total carbon buried as organic matter. Isotopic mass balance shows little difference between the average organic burial fraction for 3.6–2.5 Ga of $f_{org} = 0.15 \pm 0.02$ [2 standard deviations (SD)] and that for 1.8–1.0 Ga of $f_{org} = 0.18 \pm 0.02$ [2 SD][24], suggesting no significant increase in $O_2$ flux.

Of course, the inferred organic burial fraction and associated $O_2$ flux depend on assumptions about the carbon cycle. Challenges to a conventional carbon mass balance model include $^{12}C$-enriched carbon sequestered into seafloor carbonates[25] or into authigenic carbonates[26], or isotopic weathering inputs that scaled with the amount of atmospheric $O_2$[24,27,28]. The first two hypotheses lack supporting evidence: Archean seafloor carbonate is not isotopically lighter than typical marine sedimentary carbonates[29], and the abundance of authigenic carbonates is low before the GOE compared with afterward[30], and is relatively small today[31]. The third idea is plausible: the lack of oxidative weathering in the Archean could modulate $f_{org}$[27]. However, the average organic content of Archean sedimentary rocks (3.59 wt%) is indistinguishable from that of Proterozoic (3.56 wt%)[24], and the cumulative distribution of total organic content in organic-rich Archean sedimentary rocks is identical to that of Neogene rocks[1].

Here we are interested in examining the potential implications of the trend in mantle redox state on the GOE. A lower Archean $O_2$ flux may be a possible factor, but that is not the foxus of this paper.

Because $f_{org}$ changed little with a conventional carbon cycle model, it has been proposed that a secular decline of efficient $O_2$ sinks, rather than an increase in $f_{org}$, caused the tipping point that initiated the GOE; this point would have been reached when the efficient $O_2$ sink flux fell below the organic burial flux of $O_2$[19,22,32,33]. Models show that atmospheric $O_2$ would then rapidly accumulate, until oxidative weathering of the continents became a significant sink, causing $O_2$ concentrations to level off[34]. Proxies suggest that the $O_2$ level, however, was still far below today's atmospheric $O_2$ concentrations [e.g., ref. [35]].

Qualitatively, the pre-GOE $O_2$ sink flux could have declined if the mantle's oxidation state increased over time[32,36,37]. The proportion of reducing gases in volcanic emissions depends inversely on the oxygen fugacity ($f_{O_2}$) in their magma source region, the upper mantle. Thus, if the Archean mantle's $f_{O_2}$ was low, the $H_2/H_2O$ and $CO/CO_2$ ratios in Archean volcanic gases would have been high, suppressing atmospheric $O_2$ levels.

Until recently, studies suggested that the mantle's $f_{O_2}$ had been similar to the modern value since at least the early Archean[38–42]. The mantle $f_{O_2}$ estimated by many of these studies has uncertainties that vary between ~1 $\log_{10}$ unit[38,43] and ~2 $\log_{10}$ units[41,42]. Changes in the oxygen fugacity of the mantle as small as ~0.5 $\log_{10}$ units have been suggested to have a significant effect on atmospheric redox evolution[32]; therefore, prior results do not preclude involvement in the GOE.

A notable study of mantle redox evolution is Li and Lee[40], who used the V/Sc ratios of a large database of primitive basalts to report that mantle oxidation state had not changed by more than 0.3 $\log_{10}$ units since the Archean. The use of the V/Sc oxybarometer on large amounts of published basalt data relies upon the assumption that all of the studied basalts are the result of similar degrees of partial melting, and that they sample a spinel peridotite source with a primitive mantle V/Sc of ~5. None of these are necessarily safe assumptions. Average degree of partial melting was likely higher in the Archean due to higher mantle potential temperature. In addition, not all of the studied basalts were likely generated from melting of spinel peridotite, and residual garnet can have a potentially strong effect on V/Sc[44]. Finally, the mantle sources of basalts vary greatly in the degree of previous melt depletion, and remelting of a previously depleted source can generate a low V/Sc melt and an erroneously reduced $f_{O_2}$. For all these reasons, the conclusions and quoted uncertainties of Li and Lee[40] must come into question.

Two new studies reveal an $f_{O_2}$ trend. Aulbach and Stagno[45] carefully filtered V/Sc data to only include Archean basalts formed in a mid-ocean ridge (MOR)-like environment, while Nicklas et al.[46] calculated the $f_{O_2}$ of ultramafic lavas directly without making assumptions about their source compositions. These new data, shown in Fig. 1a, indicate that mantle $f_{O_2}$ increased by ~1.3 $\log_{10}$ units from the early Archean to Proterozoic, and likely represent the current best estimate for mantle redox evolution.

Both datasets in Fig. 1a show a similar $f_{O_2}$ trend, but each dataset was determined using a different oxybarometer. The $f_{O_2}$ derived from different oxybarometers shows a systematic offset, the reason for which is currently unclear, as oxybarometry performed using different methods on modern rocks gives values that vary outside of analytical uncertainty[47]. One possible explanation involves the degassing of volatile species, such as $SO_2$, which have the potential to strongly reduce a lava shortly prior to eruption[48]. Degassing may lead to an offset between these two datasets because the V-partitioning oxybarometer of Nicklas et al.[46] measures the $f_{O_2}$ of komatiites and picrites, while the V/Sc ratio oxybarometer of Aulbach and Stagno[45] infers the $f_{O_2}$ of basalts and picrites. Komatiites are high-temperature, high-degree

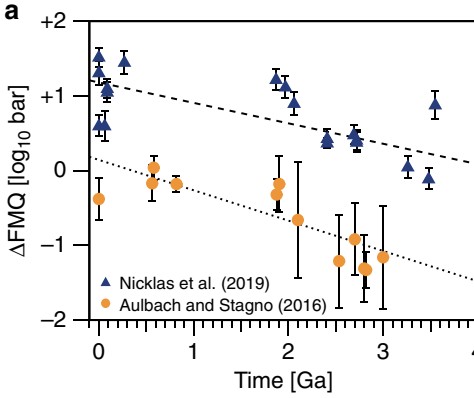

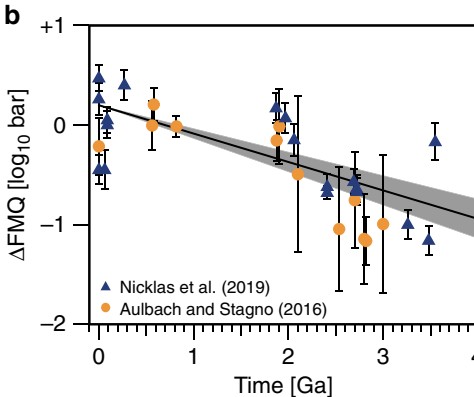

**Fig. 1 Evolution of the oxygen fugacity of mantle.** The oxygen fugacity ($f_{O_2}$) is in $\log_{10}$ units relative to the Fayalite–Magnetite–Quartz (FMQ) buffer. In **a**, we show original data of Aulbach and Stagno[45] and Nicklas et al.[46]. Dotted and dashed lines in **a** represent a linear fit for the data of Aulbach and Stagno[45] and Nicklas et al.[46], respectively, showing similar trends. Both datasets should converge on an average modern $f_{O_2}$ value inferred from mid-ocean ridge basalt (MORB). So, in **b**, we anchor the datasets to the MORB-inferred $f_{O_2}$ of the modern mantle, 0.2[50]. Thus, 0.2 is added to Aulbach and Stagno data[45] and −1.00 to Nicklas et al. data[46]. The black solid line and gray shaded region in **b** represent the median value and 95% confidence interval of oxygen fugacity, respectively. The error bar represents uncertainty of 1$\sigma$. The gray shaded region corresponds to a variation of the slope of the linear fit, which is propagated from the variations of the samples (i.e., the error bars). Note that in **b**, the variation of the $f_{O_2}$ of the modern mantle, which is discussed later, is neglected.

partial melts that are undersaturated in sulfur, while basalts are lower-degree, sulfur-saturated melts. Hence, sulfur degasses more from basaltic lavas than komatiitic lavas, i.e., decreases the $f_{O_2}$ of basalts more than that of komatiite[46].

As each mantle oxidation trend only compares samples analyzed by the same oxybarometry method, and each study also analyzed modern mid-ocean ridge basalt (MORB) with its respective method, we can anchor each trend to the current canonical modern MORB value of $+0.2 \pm 0.3 \log_{10}$ units above the fayalite–magnetite–quartz (FMQ) buffer as determined by X-ray absorption near-edge spectroscopy[49,50]. Upon anchoring, the two mantle oxidation trends overlap within their respective uncertainties, lending credence to the idea of mantle oxidation (Fig. 1b).

Although it has been speculated that the secular trends in the mantle $f_{O_2}$ in Fig. 1a and the timing of the GOE may be related[45,46], this hypothesis has never been quantitatively tested in a redox model for the surface environment using data-derived trends of $f_{org}$ and mantle $f_{O_2}$.

In this study, we show that the new data imply that reducing Archean volcanic gases would prevent atmospheric $O_2$ from accumulating, and then the GOE would occur by or after ~2.5 Ga with ≥95% probability. Thus, we conclude that secular oxidation of the mantle have indeed triggered oxygenation of the atmosphere.

## Results

**Oxygenation parameter**. We evaluate whether the atmosphere was prone to oxygenation at any given time using an oxygenation parameter, $K_{oxy}$[22,33,34]. This parameter is the ratio of $O_2$ source fluxes to kinetically efficient $O_2$ sink fluxes, which we consider here to be dominated by oxidizable volcanic gases

$$K_{oxy} \equiv \frac{O_2 \text{ source fluxes}}{\text{non-weathering } O_2 \text{ sink fluxes (may incl. excess reductants)}}. \quad (1)$$

Fluxes are quantified in units of $\text{T mol } O_2 \text{ yr}^{-1}$. When $K_{oxy} < 1$, gaseous volcanic $O_2$ sinks exceed $O_2$ sources, and excess $H_2$ builds up until balanced by escape to space. When $K_{oxy} > 1$, $O_2$ sources exceed efficient $O_2$ sinks, and $O_2$ builds up until balanced by oxidative weathering. Box modeling coupled to photochemistry shows that $K_{oxy} = 1$ defines the point when the atmosphere becomes oxic (see Fig. 7b of Claire et al.[34]). For detailed information, see the "Methods" section and the Supplementary information.

**Evolution of the oxygenation parameter**. As an initial, illustrative trial, we took the organic burial fraction ($f_{org}$) to be constant at 20%, which is a canonical value adopted by previous researchers [e.g., refs. [21,32]], to isolate the effect of the secular change in mantle $f_{O_2}$ shown in Fig. 1b. As shown in Fig. 2a, the results are that the oxygenation parameter ($K_{oxy}$) monotonically increases with time.

In Fig. 2a, $K_{oxy}$ larger than unity occurs when the production of $O_2$ via organic burial and deposition of pyrite (i.e., the numerator of Eq. (20)) exceeds the consumption of $O_2$ from oxidizable volcanic gases (i.e., the denominator of Eq. (20)). Then atmospheric $O_2$ accumulates, and oxidative weathering kicks in and balances the excess $O_2$ production[22,33,34]. In contrast, when $K_{oxy}$ is smaller than unity, the production of $O_2$ via organic burial and deposition of pyrite (i.e., the numerator of Eq. (20)) is smaller than the consumption of $O_2$ by reducing volcanic gases (i.e., the denominator of Eq. (20)). Under such a condition, atmospheric $O_2$ cannot accumulate, and the buildup of excess of reducing gases, such as $CH_4$ and $H_2$, is limited by their decomposition in the upper atmosphere and the escape of hydrogen to space[22,33,34].

The 5% probability quantile of $K_{oxy}$ (the lower end of gray shaded region) crosses unity (a dotted gray line) at 2.62 Ga in the purely illustrative case of Fig. 2a. This means that at 2.62 Ga and later, the probability that $K_{oxy}$ is larger than unity, allowing atmospheric oxygenation, is 95% or more. Hereafter, we designate this 95% threshold as the oxic transition time.

In Eq. (20), $K_{oxy}$ also depends on $f_{org}$, which has fluctuated over time. In Fig. 2b, which we subsequently call our standard case, we included temporal changes in $f_{org}$ from Krissansen-Totton et al.[24] derived from the carbon isotope record (Supplementary Fig. 1). Unlike in Fig. 2a, $K_{oxy}$ fluctuates in Fig. 2b because of fluctuations in $f_{org}$ derived from the carbon isotope record. However, $K_{oxy}$ still increases with time. Here, the oxic transition time of this standard case is 2.48 Ga, which is delayed because of relatively low values of $f_{org}$ before 2.5 Ga (Supplementary Fig. 1). Hence, even when we include temporal changes of $f_{org}$, the atmosphere still becomes oxic at 2.48 Ga (or afterward) with 95% (or more) probability.

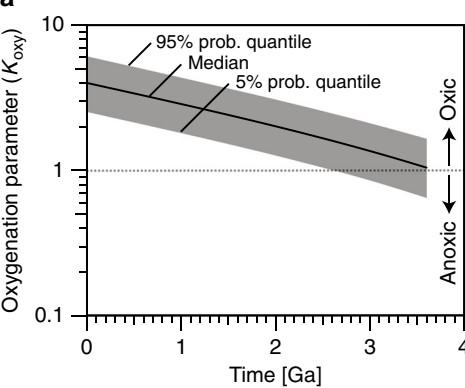

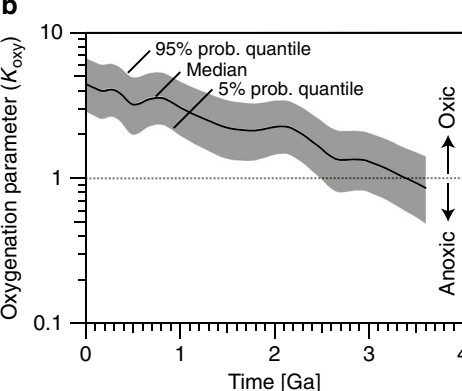

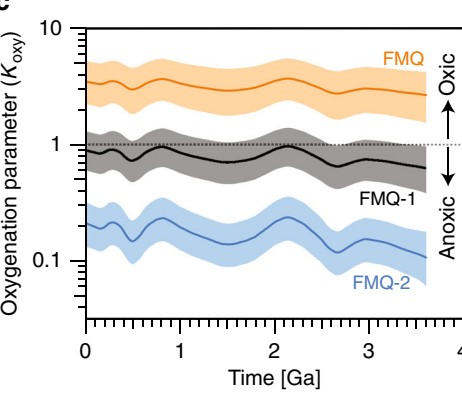

**Fig. 2 Evolution of the dimensionless oxygenation parameter, $K_{oxy}$.** Solid lines represent median values, and the shaded region bounds 5% to 95% probability quantiles. These are obtained by 10,000 times Monte-Carlo simulations. Gray dotted lines highlight $K_{oxy} = 1$, above which the atmosphere is oxic; otherwise, it is anoxic. **a** Organic burial fraction is constant at 20%, and the secular increase in the oxygen fugacity of mantle in Fig. 1b is considered. **b** Both a change in organic burial fraction from Krissansen-Totton et al.[24] and secular increase in the oxygen fugacity in Fig. 1b are considered. **c** Oxygen fugacity is assumed to be constant at three different levels (blue: FMQ-2, black: FMQ-1, orange: FMQ). The fluctuations come from imposed changes in organic burial fraction. In **b**, the parameter $K_{oxy}$ exceeds unity by ~2.5 Ga with >= 95% probability.

On the other hand, the probability of $K_{oxy} > 1$ is less than 50% before 3.38 Ga, for example, 30% at 3.6 Ga (Fig. 2b, see also Supplementary Fig. 2). This indicates that the atmosphere was likely reducing in the early Archean despite the possible presence of oxygenic photosynthesis. Thus, the increase in $f_{O_2}$ of the mantle would cause the atmosphere to shift from reducing to oxic, and the shift seemed to occur before ~2.5 Ga.

For comparison, we also did calculations for three hypothetical scenarios where the mantle $f_{O_2}$ was fixed at FMQ-2, FMQ-1, and FMQ (Fig. 2c). We also included temporal fluctuations in $f_{org}$, as in Fig. 2b, which are reflected in the fluctuations of $K_{oxy}$. In these scenarios, an oxic transition for $K_{oxy}$ does not occur, although $K_{oxy}$ occasionally becomes greater than 1 in the FMQ-1 case within the gray shaded region of uncertainty in Fig. 2c. Thus, the transitions from $K_{oxy} < 1$ to $K_{oxy} > 1$ in Fig. 2a, b are mainly due to the temporal increase in mantle $f_{O_2}$ taken from the fit in Fig. 1.

## Discussion

As shown above, an empirically-inferred increase in mantle $f_{O_2}$ causes an increase in $K_{oxy}$, resulting in the shift of the atmosphere from reducing to oxic (Fig. 2). However, this result does not exclude a role for other processes in the oxidation of the atmosphere as explained below.

The result of the standard case shows that uncertainties in the $f_{O_2}$ time series and the oxygen flux estimates from the carbon isotope record propagate through the calculations, so that the uncertainty envelope allows an oxic regime even before ~2.5 Ga (Fig. 2b). On the other hand, geological evidence, such as mass-independent sulfur isotope fractionation[1], indicates an anoxic atmosphere for the Archean Earth before 2.4–2.3 Ga though a recent study suggests that the GOE occured somewhat earlier than 2.4 Ga[51]. This discrepancy in the timing of the GOE might also be caused by not considering other processes that may delay the oxic transition.

To investigate the effect of processes proposed by previous studies[20–22,27,52,53], we did some sensitivity studies and obtained the same trends as previous studies. As explained in Supplementary Note 1, $K_{oxy}$ decreases if the degassing pressure is high, if the carbon and/or sulfur degassing fluxes are low, and/or if the rate of magnetite deposition via serpentinization is high, which generates oxidizable hydrogen that is a sink for $O_2$.

If $K_{oxy}$ was lower, the atmosphere in the early Archean would be more reducing, and the oxic transition time would be delayed, because of the following processes: a secular decrease in the degassing pressure due to a transition from submarine to subaerial volcanism [e.g., refs. [20,52]], a secular increase in the carbon and/or sulfur degassing due to an increase in their continent and/or ocean floor reservoir[21], and/or a secular decrease in the magnetite deposition flux via serpentinization, which might result from a decrease in the degree of partial melting of the mantle caused by secular cooling[22]. The magnitude of the Archean serpentinization flux of $H_2$ is debated, since it is only significant today from slow-spreading centers where ultramafic rocks are exposed[53].

There is another possible process that contributed to the delay in th oxic transition along with slow mantle oxidation. Because of the lack of oxidative weathering in the Archean, the carbon isotope input into the atmosphere-ocean could have been relatively heavy compared with mantle values, and so less organic burial was needed for the mass balance[27].

We also investigated the uncertainty of the anchoring value of the mantle $f_{O_2}$ evolution (Supplementary Note 1). For the standard case (Fig. 2), we modeled the evolution of the mantle $f_{O_2}$ anchoring the trend to the average $f_{O_2}$ of modern MORB, i.e., $\Delta FMQ_0 = +0.2$ [see also Eq. (4)][50]. However, this reference value has an uncertainty of 0.3 $\log_{10}$ units[50]. Anchoring the $f_{O_2}$ trend to a lower $\Delta FMQ_0$ value causes lower mantle $f_{O_2}$ in the past, resulting in a more reducing atmosphere in the Archean and delaying the oxic transition time (Supplementary Fig. 8).

Thus, all of the processes discussed in the preceding three paragraphs might affect the GOE's timing and so are potentially complicationg factors.

Observations show that erupted volcanic gas is more oxidized, i.e., has a larger $\Delta$FMQ, than its source melt[54,55]. The relative oxidative state of the volcanic gas results from reactions within a closed gas mixture due to cooling[54,55]. Consequently, it has been proposed that secular cooling of the mantle could have facilitated the GOE[55]. Also, observations seem to indicate that it is not possible to calculate the oxidation effect, i.e., $K_{oxy}$, of the volcanic gas using the $\Delta$FMQ of its source, although we did this in this study. However, the assumptions of these previous studies[54,55] need to be reconsidered, as follows.

Recent work[56] discusses how cooling affects the oxidation state and the $K_{oxy}$ of a volcanic gas mixture considering two stages of degassing, i.e., the stage where volcanic gas is buffered by its source melt (melt-buffered stage), and the stage where the volcanic gas is a closed system (closed stage). For the closed stage, the same conclusion was reached as in the previous studies[54,55], i.e., cooling increases $\Delta$FMQ of a closed volcanic gas compared with that of its source melt. However, in a closed gas mixture, oxidation of a gas should be accompanied by a reduction of another gas, so any reaction in the closed gas mixture does not change the overall $O_2$ sink in the gas mixture[56]. Hence, to evaluate the $K_{oxy}$ of volcanic gas, we can neglect the effect of reactions after the volcanic gas separated from its source, i.e., the observed difference in the oxidation state between erupted volcanic gas and its source melt.

For the melt-buffered stage, cooling results in reduction of a gas mixture and a decrease in $K_{oxy}$ if the $\Delta$FMQ of the source melt is buffered and constant[56]. This is because cooling decreases the absolute value of $f_{O_2}$ of the FMQ buffer (Supplementary Fig. 4). A trend of smaller $K_{oxy}$ with lower mantle temperature is shown in Supplementary Fig. 3a. However, the oxic transition time is insensitive to temperature if the temperature is higher than the solidus temperature of dry peridotite (~1390 K)[57], as shown in Supplementary Fig. 3b. The mantle temperature in the Archean would be between 1600 and 1900 K [e.g., refs. [57–59]]. Hence, the secular cooling of the mantle would affect less the GOE.

Note that Holland's $f$ number [e.g., ref. [32]] was used in Moussallam et al.[48], while $K_{oxy}$ was used in Kadoya et al.[56]. However, the temperature dependence of Holland's $f$ number is essentially the same as that of $K_{oxy}$. This point is explained in Supplementary Note 2.

A decrease in mantle temperature might delay the oxic transition time if the temperature was lower than the solidus temperature of dry peridotite (Supplementary Fig. 3b). Degassing would occur under such a low temperature in arc volcanism because the hydrous phase of the subducted crust can lower the melting temperature [e.g., ref. [60]]. However, the island arc basalts are often more oxidized than the mid-ocean ridge basalts or oceanic island basalts[61]. Hence, we cannot conclude that arc volcanism, whose degassing temperature would be low, delayed the oxic transition time.

A question might arise about our assumption that the $f_{O_2}$ of the volcanic gas was equal to that of the upper mantle. Degassing can reduce the source melt, e.g., $SO_2$ degassing[48]. However, during the melt-buffer stage, i.e., when the volcanic gas mixture interacts with the ambient melt, the $f_{O_2}$ of the gas will be equal to that of the melt. In addition, during the closed stage, i.e., after the gas mixture decouples from the ambient melt, any reaction within the gas mixture does not change the oxygenation effect of the gas[56]. Thus, the oxygenation effect of the gas should be examined using the $f_{O_2}$ of the gas when the gas decouples from the melt. Also, at that time, the $f_{O_2}$ of the gas would equal to that of the ambient melt, which has already experienced degassing. Therefore, we can calculate $K_{oxy}$ using the $f_{O_2}$ from the Archean that is anchored today to modern MORB $f_{O_2}$.

It is also noteworthy that the similar trend of the mantle $f_{O_2}$ with time is observed by the measurement of $f_{O_2}$ of different rocks[46,59]. This similarity indicates that the increase in $f_{O_2}$ of melts follows the same trend of the decreasing mantle $f_{O_2}$ with time, despite the difference in the degree of partial melting, which also supports our assumption that the $f_{O_2}$ of the volcanic gas increased with time.

The assumption of the anchoring value of the $f_{O_2}$ bears consideration. In this study, we anchored the $f_{O_2}$ of the volcanic gas and the melt at the value of the modern MORB, implicitly assuming that the $f_{O_2}$ of the melt does not change after the melt decouples from the gas. However, if the $f_{O_2}$ of the melt changes after the decoupling from the gas, it also changes the timing of the GOE as indicated by the parameter study of the anchoring valur of the $f_{O_2}$ (Supplementary Fig. 8).

Of course, a major question is what drove the increase in mantle $f_{O_2}$ and, hence, could have driven the GOE. One possibility is that convection-driven homogenization of an initially redox-stratified primordial mantle was responsible for this change[45,46,62]. The basic idea is that in the early deep mantle, $Fe^{2+}$ disproportionated to $Fe^{3+}$ and Fe metal due to high pressure. The latter was lost to the core, leaving a more oxidized lower mantle below a relatively reduced upper mantle. According to Andrault et al.[62], the primitive mantle contained excess of $Fe^{3+}$ corresponding to ~60% of an ocean's worth of oxygen.

However, another possible driver of upper mantle oxidation has recently gained evidence in its favor. Ancient air dissolved in inclusions of seawater in Archean quartz shows that the nine isotopes of Xe become increasingly isotopically heavy throughout the Archean and early Proterozoic until the GOE[63]. The most plausible explanation is a very rapid escape of hydrogen to space that dragged along ionized Xe atoms, which would have fractionated Xe isotopes because of mass-dependent escape[64].

Substantial loss of a strong reducing agent, i.e., hydrogen, would have oxidized the Earth, with the oxidation affecting the reservoir from which the hydrogen originated. Thus, the mantle would become gradually more oxidized because the hydrogen comes from the decomposition of water in volcanic melts, schematically represented as $3FeO + H_2O \rightarrow Fe_3O_4 + H_2$. The upper mantle (down to ~660-km depth) contains the equivalent of ~20% of an ocean's worth of oxygen as $Fe^{3+}$ [ref. [65], p. 207]. The Xe isotope data require Archean hydrogen loss from the equivalent of ~10 s of percent of an ocean[64]. Thus, if the $f_{O_2}$ trend in Fig. 1b explains the GOE, as shown in Fig. 2b, the $f_{O_2}$ trend, in turn, may have been driven by hydrogen escaping to space from the Earth's pre-GOE anoxic atmosphere [e.g., refs. [19,66]].

Hydrogen escape as a mechanism for mantle oxidation[36] has been proposed previously, but has been rejected over the last two decades because of reports of seemingly constant mantle $f_{O_2}$ through time[39,67,68]. Our results suggest that this mechanism may need to be reconsidered.

In summary, we examined whether new data for increasing mantle oxygen fugacity ($f_{O_2}$) since the Archean could explain the GOE, when $O_2$ first accumulated in the Earth's atmosphere. The onset of the GOE can only be properly quantified by considering sources and sinks of oxygen in a global redox balance of the surface environment. The oxygenation parameter $K_{oxy}$ used for this purpose is defined as the ratio of $O_2$ sources to kinetically rapid sinks. For an anoxic atmosphere, $K_{oxy} < 1$, while for an oxic atmosphere, $K_{oxy} > 1$; by evaluating when $K_{oxy} = 1$, we determined how mantle $f_{O_2}$ trends affected the onset of the GOE.

A more reducing mantle with low $f_{O_2}$ produces a greater proportion of reducing volcanic gases. So, we found that the data-derived trend of mantle $f_{O_2}$ likely prevented $O_2$ building up in the atmosphere with relatively high probability (~70% at 3.6 Ga) and

caused an oxic transition from $K_{oxy} < 1$ to $K_{oxy} > 1$ with >95% probability after 2.5 Ga.

Our calculated timing of the GOE is relatively insensitive to the mantle potential temperature, but depends on the assumed degassing pressure, and total outgassing fluxes of carbon and sulfur relative to total hydrogen. An additional oxidative sink of ferric iron in magnetite deposition has a minor effect, unless this flux in the Archean exceeded ten times the modern flux.

If a trend in mantle $f_{O_2}$ controlled the timing of the GOE, then the cause of mantle oxidation is ultimately important for setting the tempo of biological evolution because macroscopic, energy-intensive aerobic life was impossible when $O_2$ levels were negligible. Possible drivers of mantle redox evolution are the mixing of a redox heterogeneous mantle or the time-integrated oxidative effect of the breakdown of mantle water in volcanism, and the escape of hydrogen to space. Such processes could also apply to other Earth-like planets elsewhere, and would thus determine whether such planets could be habitats for complex aerobic life with high $O_2$ demand[69].

## Methods
The redox tipping point of the atmosphere can only be quantified by considering the global redox flux balance of the early atmosphere and ocean, which is as fundamental as mass or energy conservation[22,65, p. 221–223].

To evaluate the redox tipping point of the atmosphere, we use the oxygenation parameter, $K_{oxy}$, which is the ratio of the $O_2$ source fluxes ($F_{oxi}$) to non-weathering $O_2$ sink fluxes ($F_{red}$)[22,34,69]:

$$K_{oxy} \equiv \frac{F_{oxi}}{F_{red}}. \tag{2}$$

The fluxes, $F_{oxi}$ and $F_{red}$, are calculated using fluxes of volcanic gas, such as CO and $SO_2$, organic burial flux, and pyrite burial flux. In the subsequent sections, we will describe a model of each flux, and then explain models of $F_{oxi}$ and $F_{red}$.

**Fluxes related to source and sink of oxygen.** In this section, we explain the components, which are used to calculate the $O_2$ source ($F_{oxi}$) and a kinetically rapid sink of $O_2$ ($F_{red}$). For the calculation of $F_{oxi}$ and $F_{red}$, see the next section.

Hydrogen is degassed to the ocean–atmosphere system as hydrogen molecules ($H_2$), water vapor ($H_2O$), methane ($CH_4$), and hydrogen sulfide ($H_2S$). Accordingly, a total flux of hydrogen ($F_{hydrogen}$) can be expressed as follows:

$$F_{hydrogen} = F_{H_2} + F_{H_2O} + 2F_{CH_4} + F_{H_2S}, \tag{3}$$

where $F_x$ is a flux of $x$. Methane contains the equivalent of two $H_2$ molecules, so $F_{CH_4}$ is weighted by a factor of 2.

Carbon is degassed to the ocean–atmosphere system as carbon dioxide ($CO_2$), carbon monoxide (CO), and methane ($CH_4$), and deposited as organic matter (org) and carbonate (carb). Since we assume that carbon is in a steady state, the total flux of carbon ($F_{carbon}$) is equal to input and output fluxes, which can be expressed as follows:

$$F_{carbon} \equiv F_{CO_2} + F_{CO} + F_{CH_4} = F_{org} + F_{carb}, \tag{4}$$

where $F_x$ is a flux of $x$ as in Eq. (3).

The ratio of the organic burial to the total carbon flux (i.e., $f_{org} = F_{carb}/F_{carbon}$) can be evaluated using the geological record of carbon isotopes in organic and inorganic carbon [e.g., ref. [24]]. Supplementary Fig. 1 shows the temporal change in the $f_{org}$, which is derived by Krissansen-Totton et al.[24].

Similarly, sulfur (S) is degassed to the ocean–atmosphere system as sulfur dioxide ($SO_2$) and hydrogen sulfide ($H_2S$), and deposited as pyrite ($FeS_2$) and sulfate (e.g., $CaSO_4$). However, we neglect the deposition of sulfate because we focus on the Archean eon, the surface environment was anoxic, and there was very little sulfate (~2.5 μM, i.e., 0.01% of modern level) in the Archean ocean[70,71]. Since we assume that sulfur is in steady state, a total flux of sulfur ($F_{sulfur}$) is equal to input and output fluxes, which can be expressed as follows:

$$F_{sulfur} \equiv F_{SO_2} + F_{H_2S} = \frac{1}{2}F_{FeS_2}, \tag{5}$$

where $F_x$ is a flux of $x$ as in Eq. (3).

**Sources and sinks of oxygen.** In this section, we explain models of the $O_2$ source ($F_{oxi}$) and a kinetically rapid sink of $O_2$ ($F_{red}$), which are used for the calculation of $K_{oxy}$ (Eq. (2)).

We take $H_2O$ and $CO_2$ to be redox-neutral, so hydrogen ($H_2$), carbon monoxide (CO), and methane ($CH_4$) are sinks of oxygen, as follows:

$$H_2 + \frac{1}{2}O_2 \rightarrow H_2O, \tag{6}$$

$$CO + \frac{1}{2}O_2 \rightarrow CO_2, \tag{7}$$

$$CH_4 + 2O_2 \rightarrow CO_2 + 2H_2O. \tag{8}$$

Hence, one mole of $H_2$ degassing corresponds to 0.5 mole of $O_2$ consumption. Similarly, one mole of CO and $CH_4$ degassing corresponds to 0.5 and 2 moles of $O_2$ consumption, respectively.

On the other hand, the burial of organic matter ($CH_2O$) is a source of oxygen, as follows:

$$CO_2 + H_2O \rightarrow CH_2O + O_2. \tag{9}$$

Hence, one mole of the burial of organic matter corresponds to one mole of $O_2$ production.

We also assume that $SO_2$ is redox-neutral, so $H_2S$ is a sink of oxygen, as follows:

$$H_2S + \frac{3}{2}O_2 \rightarrow SO_2 + H_2O. \tag{10}$$

Also, reduction of $SO_2$ must work as a source of oxygen, as follows:

$$SO_2 + \frac{1}{2}FeO \rightarrow \frac{1}{2}FeS_2 + \frac{5}{4}O_2. \tag{11}$$

Hence, one mole of $SO_2$ degassing corresponds to 1.25 mole of $O_2$ production. On the other hand, one mole of $H_2S$ degassing corresponds to 0.25 mole of $O_2$ consumption.

As explained above, degassing of $H_2$, CO, $CH_4$, and $H_2S$ is a sink of oxygen. Accordingly, $F_{red}$ is expressed as follows:

$$F_{red} = \frac{1}{2}F_{H_2} + \frac{1}{2}F_{CO} + 2F_{CH_4} + \frac{1}{4}F_{H_2S}. \tag{12}$$

On the other hand, burial of organic matter and $SO_2$ degassing are sources of oxygen. Accordingly, $F_{oxi}$ is expressed as follows:

$$F_{oxi} = F_{org} + \frac{5}{4}F_{SO_2}. \tag{13}$$

**Derivation of oxygenation parameter.** Substituting Eqs. (12) and (13) into Eq. (2), $K_{oxy}$ is rewritten as follows:

$$K_{oxy} = \frac{4F_{org} + 5F_{SO_2}}{2F_{H_2} + 2F_{CO} + 8F_{CH_4} + F_{H_2S}}. \tag{14}$$

We now define $\chi_c$ and $\chi_s$ as follows:

$$\chi_c \equiv \frac{F_{carbon}}{F_{hydrogen}}, \chi_s \equiv \frac{F_{sulfur}}{F_{hydrogen}}. \tag{15}$$

In addition, we define $r_{H_2}$ as the ratio of a flux of hydrogen molecule to a total flux of hydrogen:

$$r_{H_2} \equiv \frac{F_{H_2}}{F_{hydrogen}}. \tag{16}$$

Similarly, we defined the following parameters:

$$r_{CO} \equiv \frac{F_{CO}}{F_{carbon}}, r_{CH_4} \equiv \frac{F_{CH_4}}{F_{carbon}}, f_{org} \equiv \frac{F_{org}}{F_{carbon}}, \tag{17}$$

$$r_{SO_2} \equiv \frac{F_{SO_2}}{F_{sulfur}}, r_{H_2S} \equiv \frac{F_{H_2S}}{F_{sulfur}}. \tag{18}$$

If we substitute from the above for the various gas fluxes and flux of organic burial, Eq. (14) can be rewritten as

$$K_{oxy} = \frac{4f_{org}F_{carbon} + 5r_{SO_2}F_{sulfur}}{2r_{H_2}F_{hydrogen} + (2r_{CO} + 8r_{CH_4})F_{carbon} + r_{H_2S}F_{sulfur}}. \tag{19}$$

If we now divide the above equation by $F_{hydrogen}$, we arrive at the form

$$K_{oxy} = \frac{4f_{org}\chi_c + 5r_{SO_2}\chi_s}{2r_{H_2} + (2r_{CO} + 8r_{CH_4})\chi_c + r_{H_2S}\chi_s}. \tag{20}$$

Kasting[22] argues that an additional $O_2$ flux, in the form of $F_{Fe_3O_4}$, could be included in the denominator of Eq. (2). If we take FeO as the redox reference state of iron for the Archean surface environment, then the formation of ferric iron in magnetite ($Fe^{3+}_2Fe^{2+}O_4$) in iron formations or during serpentinization and its burial removes oxidizing power from the surface environment or, equivalently, is an input flux of reducing power. We initially neglect the deposition of magnetite, and then examine its specific influence later. When the magnetite deposition is

taken into account, $K_{oxy}$ can be rewritten as

$$K_{oxy} = \frac{4f_{org}\chi_c + 5r_{SO_2}\chi_s}{2r_{H_2} + \left(2r_{CO} + 8r_{CH_4}\right)\chi_c + r_{H_2S}\chi_s + \frac{F_{Fe_3O_4}}{F_{hydrogen}}},$$ (21)

where $F_{Fe_3O_4}$ is an $O_2$ consumption via magnetite deposition in the unit of T mol $O_2$ yr$^{-1}$. For example, the modern $F_{Fe_3O_4}$ is $0.05 \sim 0.2$ T mol $O_2$ yr$^{-1}$ [72,73].

**Equilibrium state of volcanic gases**. In this section, we will explain how to calculate the flux ratios, i.e., $r_x$ for each volatile species $x$ in Eq. (20).

To calculate fluxes of volcanic gases, we first assume that equilibrium states are achieved for volcanic volatiles in the silicate melt:

$$\begin{cases} H_2O = H_2 + \frac{1}{2}O_2 \\ CO_2 = CO + \frac{1}{2}O_2 \\ CO_2 + 2H_2O = CH_4 + 2O_2 \\ SO_2 + H_2O = H_2S + \frac{3}{2}O_2 \end{cases}.$$ (22)

These equations have equilibrium constants that are as follows, in terms of fugacities ($f_x$) for each volatile species $x$:

$$\begin{cases} K_1 = \frac{f_{H_2}f_{O_2}^{0.5}}{f_{H_2O}} \\ K_2 = \frac{f_{CO}f_{O_2}^{0.5}}{f_{CO_2}} \\ K_3 = \frac{f_{CH_4}f_{O_2}^2}{f_{CO_2}f_{H_2O}^2} \\ K_4 = \frac{f_{H_2S}f_{O_2}^{1.5}}{f_{SO_2}f_{H_2O}} \end{cases}.$$ (23)

We assume that gas fluxes will be in proportion to their fugacities, e.g., $f_{H_2} = P_{tot} \times F_{H_2}/F_{tot}$. Here, $P_{tot}$ is a total pressure under which degassing occurs. $F_{tot}$ is a total degassing flux that can be written as

$$F_{tot} \approx F_{H_2O} + F_{H_2} + F_{carbon} + F_{sulfur},$$ (24)

$$= F_{hydrogen}\left(r_{H_2O} + r_{H_2} + \chi_c + \chi_s\right).$$ (25)

Then, we obtain the following equations:

$$\begin{cases} \frac{K_1}{f_{O_2}^{0.5}} = \frac{r_{H_2}}{r_{H_2O}} \\ \frac{K_2}{f_{O_2}^{0.5}} = \frac{r_{CO}}{r_{CO_2}} \\ \frac{K_3}{f_{O_2}^2}P_{tot}^2 = \frac{r_{CH_4}}{r_{CO_2}}\left(\frac{r_{H_2O}+r_{H_2}+\chi_c+\chi_s}{r_{H_2O}}\right)^2 \\ \frac{K_4}{f_{O_2}^{1.5}}P_{tot} = \frac{r_{H_2S}}{r_{SO_2}} \times \frac{r_{H_2O}+r_{H_2}+\chi_c+\chi_s}{r_{H_2O}} \end{cases}.$$ (26)

where $f_{O_2}$ is a oxygen fugacity, which we will explain later. We calculate equilibrium constants of the above reactions using data of NIST[74]. According to Eqs. (3)–(5), we also obtain the following equations:

$$\begin{cases} r_{H_2} + r_{H2O} + 2r_{CH_4}\chi_c + r_{H_2S}\chi_s = 1 \\ r_{CO_2} + r_{CO} + r_{CH_4} = 1 \\ r_{SO_2} + r_{H_2S} = 1 \end{cases}.$$ (27)

Hence, given mantle temperature, a total pressure, and oxygen fugacity in the mantle, and solving Eqs. (26) and (27), we obtain fluxes of each molecular species.

Hereafter, we explain how to solve Eqs. (26) and (27). First of all, we defined variables as follows:

$$A \equiv \frac{K_1}{f_{O_2}^{0.5}}, B \equiv \frac{K_2}{f_{O_2}^{0.5}}, C \equiv \frac{K_3}{f_{O_2}^2}P_{tot}^2, D \equiv \frac{K_4}{f_{O_2}^{1.5}}P_{tot},$$

$$r_{tot} \equiv (1 + A)r_{H_2O} + \chi_c + \chi_s.$$

Then, Eq. (27) can be rewritten as follows:

$$\begin{cases} (1+A)r_{H_2O} + 2\chi_c C\left(\frac{r_{H_2O}}{r_{tot}}\right)^2 r_{CO_2} + \chi_s D\left(\frac{r_{H_2O}}{r_{tot}}\right)r_{SO_2} = 1 \\ r_{CO_2} + Br_{CO_2} + C\left(\frac{r_{H_2O}}{r_{tot}}\right)^2 r_{CO_2} = 1 \\ r_{SO_2} + D\left(\frac{r_{H_2O}}{r_{tot}}\right)r_{SO_2} = 1 \end{cases}.$$ (28)

Hence,

$$r_{CO_2} = \frac{r_{tot}^2}{(1+B)r_{tot}^2 + Cr_{H_2O}^2},$$ (29)

$$r_{SO_2} = \frac{r_{tot}}{r_{tot} + Dr_{H_2O}}.$$ (30)

The unknown variable, $f_{H_2O}$, is obtained by solving the following equation:

$$(1+A)r_{H_2O} + 2\chi_c\frac{Cr_{H_2O}^2}{(1+B)r_{tot}^2 + Cr_{H_2O}^2} + \chi_s\frac{Dr_{H_2O}}{r_{tot} + Dr_{H_2O}} = 1.$$ (31)

**Oxygen fugacity of mantle**. As explained above, we need the oxygen fugacity, $f_{O_2}$, of volcanic gas to calculate the volcanic gas speciation. To evaluate the $K_{oxy}$ of the gas, we can use the $f_{O_2}$ of the source of the gas, i.e., the mantle melt[56]. In addition, both experimental[75] and empirical data from natural samples[76] show that the mantle melt records the same $f_{O_2}$ as its mantle residue, and therefore its mantle source. Hence, we evaluate the $K_{oxy}$ using the $f_{O_2}$ of the mantle.

According to Aulbach and Stagno[45] and Nicklas et al.[46], the oxygen fugacity of the mantle has increased by $+0.9 \pm 0.2$ [2 SD] in $log_{10}$ units during the past 3 Gyr. Since two independent datasets show the same trend, secular oxidation of the mantle is corroborated. In addition, the oxygen fugacity of the modern mantle is $+0.2 \pm 0.6$ [2 SD] in $log_{10}$ units above the FMQ buffer[50]. Here, FMQ is the fayalite–magnetite–quartz synthetic buffer, which defines $f_{O_2}$ at a particular temperature and pressure, e.g., $f_{O_2} = 10^{-8.5}$ bar at 1200 °C (1473.15 K and 0.5 MPa). Hence, we modeled the evolution of the mantle $f_{O_2}$ in $log_{10}$ units above the FMQ buffer as follows:

$$\Delta FMQ = \Delta FMQ_0 + at.$$ (32)

where $t$ is time in units of Ga, and the slope, $a$, is $-0.29 \pm 0.05$ [2 SD] in units of per Ga. For the standard case, we set $\Delta FMQ_0$ at 0.2 $log_{10}$ units[50], and later discuss the effect of the variation of this parameter.

Figure 1b shows the evolutionary range of oxygen fugacity for the standard case with its uncertainties. Further details for the comparison of the datasets of Fig. 1b are given in Supplementary Note 3.

**Other input parameters**. As indicated in Eqs. (30) and (31), the flux ratios, $r_x$ in Eq. (20), depend on the total flux ratios of carbon and sulfur relative to hydrogen, $\chi_c$ and $\chi_s$. In addition, the flux ratios also depend on the temperature and pressure of the system.

Initially, we assume modern values for the total flux ratios of carbon and sulfur relative to hydrogen. The modern total degassing fluxes of hydrogen, carbon, and sulfur (in all their forms) are $97 \pm 20$, $9 \pm 2$, and $2.2 \pm 0.7$ T mol yr$^{-1}$, respectively [ref. [65], p. 203 and p. 221]. Hence, present-day $\chi_c = 0.1 \pm 0.03$ and $\chi_s = 0.023 \pm 0.0086$. Later, we examine the effect of these parameters on the results of our modeling.

Similarly, we use a modern mantle potential temperature of 1623.15 K (1350 °C)[77] and a pressure of 0.5 MPa[78] first. We examine the sensitivity to these parameters later.

## Data availability
The datasets generated during the current study are available in the Zenodo repository [https://doi.org/10.5281/zenodo.3668382].

## Code availability
The source code used in this study is also available in the Zenodo repository [https://doi.org/10.5281/zenodo.3668382].

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

## Acknowledgements

Funding support came from NSF Frontiers in Earth System Dynamics award No. 1338810.

## Author contributions

D.C.C. designed the study. S.K. performed the calculation and data analysis. S.K. with D.C.C. wrote the paper with input from R.W.N., I.S.P., and A.D.A. All authors discussed the results and commented on the paper.

## Competing interests

The authors declare no competing interests.
