## [Peer Review File · Nature Communications]

Reviewers' comments:

Reviewer #1 (Remarks to the Author):

In this study the authors aimed to improve the knowledge of an important and long-standing enigma regarding the link between mantle oxidation state and atmospheric composition, more specifically why the GOE appears slightly shifted with respect to the first recorded oxygen whiffs. The authors performed calculations that allow to estimated the balance between reduced gas species vs oxygenated species and related this to volcanism intended as redox carrier from the Earth's mantle to the atmosphere. By analogy, the mantle-atmosphere system can be extended to other planets and provide insights on the origin of life.

the paper is certainly well written, but any calculation performed relies on correct and sometimes wrong interpretation of data from literature whose uncertainties are not taken into account at all. Figure 1, the motivation for this study, shows an increase of f_{O_2} relative to few mantle and orogenic eclogites that are assumed to reflect the f_{O_2} of Archaean MORBs without taken into consideration possible degassing processes. Wrongly, the authors here put the mantle f_{O_2} being equal to the magma f_{O_2} and the emitted gas f_{O_2} . But this is not the case as discussed by few authors recently (Stagno et al. 2019 CUP; Sorbadere et al GCA; Mooussallam et al EPSL, etc). There is not evidence that an f_{O_2} equilibrium exists between these three phases. In addition, data from Nicklas are wrongly interpreted being the difference from Aulbach and Stagno (2016) be due to the fact that the f_{O_2} is not determined at mantle source P-T conditions. For some reason the authors apply an arbitrary correction of Nicklas data to make the conciding with the present-day MORB f_{O_2} . But, again, Aulbacj and Stagno results can merely be extended to the whole mantle. Those are few sparse data. When extrapolating to the entire globe, things become highly speculative. Again, it is not clear how uncertainties derived from f_{O_2} , gas chemistry, mantle P-T regime, atmospheric T, CO_2/SO_2 gas ratio ultimately affect this model.

However, i admit that the paper is precious in defining processes of oxidation and reduction through gas/mineral reactions even better than done so far. But, quantities are not well constrained. So, I am sorry to not give the expected credit to this work. It is a great excercise, but, as said, highly speculative.

Best regards

Reviewer #2 (Remarks to the Author):

This is an interesting contribution on the causes of oxygenation of Earth's atmosphere. The paper is very clearly written, listing and explaining every step of the calculations performed as well as the assumptions behind the equations. The authors show how every parameter affects oxygen production, under the fundamental assumption that what is the ultimate driver or responsible of atmosphere redox change is the secular oxidation of the mantle. I think this is very good illustrative and informative paper that merits being published in Nature Communications.

My main comment is the lack of consideration given to recent findings showing how volcanic gas temperature changes with time (owing to secular Earth cooling) may have affected atmosphere oxygenation. We have shown that there is a decoupling between magma redox state and that of its gases when released in open atmosphere and prior to mixing with it (Oppenheimer et al., 2018, Nature Geosciences). This observation, first made on Kilauea, has been generalised by Moussallam et al (2019, EPSL) and proposed as one (one more..) contributing factor to the GOE. In short, colder gases are more oxidised than those hotter, regardless of the source's redox state. I think that for completeness this mechanism should be quoted as well in this review type paper.

Apart from that missing information, the paper is technically ok and could be published in its present form.

Other comments are listed below keyed to line numbers of the main text.

126-129. It is true that degassing affects magma redox state, but if the Hawaiian case holds true for Archean times, then we should see the reverse, ie the Aulbachs and Stagno trend should be more oxidised than the one of Nicklas et al. In general, the evidence point to a reducing effect on melt arising from volatile release during magma decompression.

155. For detailed information ?

208. I dont think this statement is entirely correct. Serpentinisation is driven, by definition, by the alteration of olivine. Nowadays, ocean floor is covered mostly by basalt with little olivine (relative to mantle). H₂ rich fluids are produced when mantle is the altered rock, not basalt, outcropping on seafloor. During the Archean, perhaps there was more extensive melting but the question is whether the produced crust was mafic or ultramafic. If it was basalt, even a more mafic than nowadays, then its alteration by seawater circulation is unlikely to have produced significant H₂.

210. I beg to disagree with this statement. See Moussallam et al 2019, EPSL

221. This temperature is equivalent to assume that only basalts at ridges produce volcanic gases, an unlikely scenario. What about volcanic gases emitted at subduction zones, which are colder by 200-300°C ? Or do you assume that subduction was not operating on a large scale prior to GOE ? Please clarify.

226. I dont know this paper but the available evidence tells rather the opposite (See Moussallam et al., 2019)

257. Perhaps say : The oxic transition time is older (larger)...

360-361. This is what the vast majority believes which I also beg to disagree with. The fact that the redox state of magma source(s) is equal to that of the magma itself is an implicit yet debatable assumption (see Burgisser and Scaillet, 2007, Nature).

Bruno Scaillet

Reviewer #3 (Remarks to the Author):

The model presented here aims at evaluating quantitatively as to whether or not the oxidation of the mantle during the Hadean and the Archean may have caused the GOE. This is an important step forward that is appropriate for publication in Nature Communications. I have several point of criticisms that need to be addressed before the paper can be accepted for publication.

The model relies on two main variables. The temporal change in the mantle fO₂ and the organic burial fraction (f_{org}) estimated from the mass balance of the carbon isotope record. Figure 2b shows that the inferred threshold of K_{oxy} = 1 above which the atmosphere becomes permanently oxidized intercept the 95% confidence interval at about 2.5 Ga. However, this is critically dependent of the shape of the undulating curve, with an inflexion point located at K_{oxy} ~ 0.85 and ~ 2.65 Ga. Because this undulation is dependent on the temporal changes in f_{org}, this inflexion point should be linked to a change in f_{org}. As shown by Krissansen-Totton et al. (2015), however, there is no statistically significant difference between the pre-GOE and post-GOE f_{org}. Rather, there is a statistically significant increase in ε (d¹³C_{carb}-d¹³C_{org}), which is linked to the decrease in d¹³C_{org} in the late Archean, possibly due to methanotrophic-recycling. This is in contrast with the transition from the Proterozoic to the Phanerozoic, which is characterized by a well-constrained

increase in f_{org} . How do these results fit with the curve shown in Figure 2b? Were both variables (f_{org} and ϵ) implemented and tested in the calculation of K_{oxy} ? How sensitive are these 2 variables on the location of the inflexion point at ~ 2.65 Ga? In the discussion section (lines 300-312), it is stressed that the uncertainty envelope allows an oxic regime before 2.4 Ga, which was not advocated in previous studies. Can be appropriate to refer to the study of Philippot et al. (2018, Nature Communications) who recently showed that the GOE initiated at least at ~ 2.45 Ga and most likely earlier.

The notion that the GOE may be the result of convection-induced homogenization of a redox-stratified mantle inherited from the disproportionation of Fe^{2+} into a Fe^{3+} rich phase stored in the lower mantle and metallic Fe^0 drained into the core has been proposed by Andraut et al., (2017, Geochim. Persp. Lett.; missing reference to be added here). Their synchrotron-based experiments indicate a primordial Fe^{3+} excess of $\sim 20\%$ for the mantle iron, which would correspond to 500–1000 times the O_2 content in the present day Earth's atmosphere. How this estimate fits with the model presented here? Considering the quantitative approach proposed, it would be of major interest to develop on the total flux of free oxygen release to the atmosphere during the Archean-Proterozoic transition.

We thank the reviewers for continuing support to improve our manuscript and for volunteering their time. Here, we have modified our manuscript following the comments with **changes that are highlighted in blue color**. In addition, we put the explanations about the sensitivity studies in the supplementary information, briefly referring to the corresponding section in the main text. Below reviewer comments are in normal typeface, and **our responses and comments are in bold blue typeface**.

A general comment for Reviewers 1 and 2 concerning the effect of temperature on the redox state of volcanic gas mixture

Both reviewers 1 and 2 cite a paper by Moussallam et al., 2019, *EPSL*, which describes how the f_{O_2} of a volcanic gas mixture changes after it decouples from a melt and claims that this f_{O_2} shift can have a profound effect on the redox state of the atmosphere.

Unfortunately, the second claim of Moussallam et al., that the redox state of the atmosphere can be affected, makes a fundamental conceptual error – the neglect of redox conservation – as we explain below. Consequently, all criticism of our manuscript on this basis is invalid. Before submitting the present manuscript, we submitted a separate paper that has been accepted by *Geochem. Perspect. Lett.* and is attached to this reply. Here, we give a brief explanation, and have added brief discussion at line 250.

It is observed that the ΔFMQ of erupted gas is larger than that of its source region magma (e.g., Oppenheimer et al., 2018, *Nat. Geosci.*; Moussallam et al., 2019, *EPSL*) but we computed the oxygenation parameter, K_{oxy} , assuming that f_{O_2} of volcanic gas was equal to that of the source region melt because it is redox conservation that matters. The observed difference in ΔFMQ is caused by reactions within gas bubbles after the bubble becomes decoupled from the source region magma, i.e., reactions within a closed system (e.g., Oppenheimer et al., 2018; Moussallam et al., 2019). However, such reactions within the closed system of gas bubble gases cannot affect the overall O_2 sink because oxidation of one volcanic gas must be accompanied by reduction of another gas as a fundamental law of chemistry.

A gas bubble separated from its source magma ascends and cools adiabatically (e.g., Oppenheimer et al., 2018). The cooling changes equilibrium states among gas species within the gas bubble. In particular for a gas bubble containing H, C, and S, the cooling causes oxidation of H_2 to H_2O and of CO to CO_2 , and reduction of SO_2 to H_2S (e.g., Moussallam et al., 2019). Also, such oxidation of H_2 and CO causes an increase in apparent f_{O_2} of the gas bubble, which can be computed from ratios of H_2O / H_2 and/or CO_2 / CO (e.g., Moussallam et al., 2019).

However, any reaction within a closed system cannot change the overall O_2 sink within the system. Consider a simple but illustrative example: the complete oxidation of an initial bubble mixture of 3 moles of H_2 and 1 mole of SO_2 , i.e., $3H_2 + SO_2 \rightarrow 2H_2O + H_2S$, which is a typical reaction for a volcanic gas bubble as mentioned above. The initial O_2 sink from 3 moles of H_2 and 1 mole of SO_2 is 0.25 mol O_2 , which is 1.5 mol O_2 sink (from H_2) minus a 1.25 mol O_2 source from SO_2 (using stoichiometric redox conventions explained in Section S1.2, eqs. S.4 and

S.9). Once the reaction goes to completion the H₂O is redox neutral and the 1 mole of H₂S is a net sink of 0.25 mol O₂ (eqs. S.8+S.9). Thus, we start with a 0.25 mol O₂ sink and end with a 0.25 mol O₂ sink, even though a computed f_{O_2} for the closed system may change.

The bottom line is that the atmospheric oxidation effect of a gas mixture does *not* depend on a change of f_{O_2} of the gas mixture that results from closed system reactions after the gas decouples from its source. Instead, atmospheric oxidation depends on the *initial* f_{O_2} of the gas mixture when the gas decouples with its source. Before the gas decouples with its source, it is plausible that the f_{O_2} of the gas is equal to that of its source because the gas components are in equilibrium with the source.

Reviewer #1

In this study the authors aimed to improve the knowledge of an important and long-standing enigma regarding the link between mantle oxidation state and atmospheric composition, more specifically why the GOE appears slightly shifted with respect to the first recorded oxygen whiffs. The authors performed calculations that allow to estimated the balance between reduced gas species vs oxygenated species and related this to volcanism intended as redox carrier from the Earth's mantle to the atmosphere. By analogy, the mantle-atmosphere system can be extended to other planets and provide insights on the origin of life.

the paper is certainly well written, but any calculation performed relies on correct and sometimes wrong interpretation of data from literature whose uncertainties are not taken into account at all.

Comment 1.1:

Figure 1, the motivation for this study, shows an increase of f_{O_2} relative to few mantle and orogenic eclogites that are assumed to reflect the f_{O_2} of Archaean MORBs without taken into consideration possible degassing processes. Wrongly, the authors here put the mantle f_{O_2} being equal to the magma f_{O_2} and the emitted gas f_{O_2} . But this is not the case as discussed by few authors recently (Stagno et al. 2019 *CUP*; Sorbadere et al *GCA*; Mooussallam et al *EPSL*, etc). There is not evidence that an f_{O_2} equilibrium exists between these three phases.

As explained in our general comment above, any reaction after a gas bubble decouples with its source magma does not change the overall O_2 sink of the bubble even though the reaction does change gas composition and f_{O_2} of the gas bubble. Hence, the observation that shows difference in computed f_{O_2} of an erupted gas bubble that cools compared to a source magma is irrelevant to considerations about the overall O_2 sink in the atmosphere, as explained in our general comment above.

Additionally, it has been shown using both experimental data (Davis and Cottrell, 2018) and empirical data from natural samples (Birner et al., 2018) that mantle melts record the same oxygen fugacity as their mantle residues; as a result, mantle melts reflect f_{O_2} of their mantle sources and, thus, can be used to study the evolution of mantle oxygen fugacity. Further, as discussed in Nicklas et al. (2019), there is no a priori reason why older lavas should have experienced larger extents of S degassing and, thus, have lower f_{O_2} , than modern lavas. It is highly unlikely that both datasets, by happenstance, included only older and more degassed lavas and younger less degassed lavas. Thus, we conclude that source f_{O_2} variation is the culprit for the trends observed.

We added a paragraph to explain the f_{O_2} equilibrium between volcanic gas, its source melt, and the mantle at line 410.

Comment 1.2:

In addition, data from Nicklas are wrongly interpreted being the difference from Aulbach and Stagno (2016) be due to the fact that the f_{O_2} is not determined at mantle source P-T conditions. For some reason the authors apply an arbitrary correction of Nicklas data to make the coinciding with the present-day MORB f_{O_2} . But, again, Aulbach and Stagno results can merely be extended to the whole mantle. Those are few sparse data. When extrapolating to the entire globe, things

become highly speculative. Again, it is not clear how uncertainties derived from f_{O_2} , gas chemistry, mantle P-T regime, atmospheric T, CO_2/SO_2 gas ratio ultimately affect this model.

Offsets between different igneous oxybarometers are by now well documented (e.g., Mallmann and O'Neill, 2013). Work using different oxybarometers can correct for offsets by analyzing some of the same samples, in this case MORB. Even if the offset between the two datasets is not constant, it is highly unlikely that different datasets would show a trend of such similar magnitude. Furthermore, even though both the Aulbach and Stagno (2016) and Nicklas et al. (2019) datasets are relatively small, they are the most relevant, high-precision f_{O_2} data currently available for samples across the Archean-Proterozoic boundary. Again, it is highly unlikely that both datasets would show a significant oxidation trend of the same magnitude in the same timeframe without this being a global feature. Conclusions in this paper are of course subject to change as new data are acquired, but the authors feel that two separate independent studies with the same conclusions warrant the modeling performed in this paper. Uncertainties on both the presented model and f_{O_2} datasets have been addressed both in the manuscript and in the two cited publications.

However, I admit that the paper is precious in defining processes of oxidation and reduction through gas/mineral reactions even better than done so far. But, quantities are not well constrained. So, I am sorry to not give the expected credit to this work. It is a great exercise, but, as said, highly speculative.

Best regards

Reviewer #2

This is an interesting contribution on the causes of oxygenation of Earth's atmosphere. The paper is very clearly written, listing and explaining every step of the calculations performed as well as the assumptions behind the equations. The authors show how every parameter affects oxygen production, under the fundamental assumption that what is the ultimate driver or responsible of atmosphere redox change is the secular oxidation of the mantle. I think this is very good illustrative and informative paper that merits being published in Nature Communications.

Comment 2.1:

My main comment is the lack of consideration given to recent findings showing how volcanic gas temperature changes with time (owing to secular Earth cooling) may have affected atmosphere oxygenation. We have shown that there is a decoupling between magma redox state and that of its gases when released in open atmosphere and prior to mixing with it (Oppenheimer et al., 2018, *Nature Geosciences*). This observation, first made on Kilauea, has been generalised by Moussallam et al (2019, *EPSL*) and proposed as one (one more..) contributing factor to the GOE. In short, colder gases are more oxidised than those hotter, regardless of the source's redox state. I think that for completeness this mechanism should be quoted as well in this review type paper.

Apart from that missing information, the paper is technically ok and could be published in its present form.

In brief, Moussallam et al. make an incorrect argument that violates redox conservation when they claim that the computed temperature-dependent f_{O_2} change in a closed system bubble of gas can affect the redox state of the atmosphere. This is fully explained in our "general comment" above. Since this point is related to Comment 2.7, we also collectively state a response later under Comment 2.7.

Other comments are listed below keyed to line numbers of the main text.

Comment 2.2:

126--129. It is true that degassing affects magma redox state, but if the Hawaiian case holds true for Archean times, then we should see the reverse, ie the Aulbachs and Stagno trend should be more oxidised than the one of Nicklas et al. In general, the evidence point to a reducing effect on melt arising from volatile release during magma decompression.

The offset between the Aulbach and Stagno (2016) and Nicklas et al. (2019) datasets is indeed consistent, with possibly greater amounts of degassing in the more evolved samples analyzed by Aulbach and Stagno (2016). As basalts are more evolved melts than komatiites, they are more likely to have degassed significant amount of sulfur and are, thus, more reduced. As sulfur is present in melts as S^{-2} and degasses dominantly as SO_2 or H_2S , degassing can only have a reducing or net zero effect, respectively. See references Nicklas et al. (2019) and Moussallam et al. (2016) for more discussion of the effect of degassing on magmatic oxygen fugacity.

We reworted the sentence at line 125 adding an explanation at line 129.

Comment 2.3:

155. For detailed information?

Done: added at line 157 in the revised manuscript.

Comment 2.4:

208. I don't think this statement is entirely correct. Serpentinisation is driven, by definition, by the alteration of olivine. Nowadays, ocean floor is covered mostly by basalt with little olivine (relative to mantle). H₂ rich fluids are produced when mantle is the altered rock, not basalt, outcropping on seafloor. During the Archean, perhaps there was more extensive melting but the question is whether the produced crust was mafic or ultramafic. If it was basalt, even a more mafic than nowadays, then its alteration by seawater circulation is unlikely to have produced significant H₂.

In this section, we considered a probability of higher serpentinization rate which resulted from higher olivine content in oceanic crust. To highlight this point, we reworded the sentences at 230 in the main text, and lines 296 and 297 in the supplementary information.

Comment 2.5:

210. I beg to disagree with this statement. See Moussallam et al 2019, *EPSL*

Line 210: "As explained below, the mantle temperature has little effect on the oxic transition time."

Unfortunately, the opinion expressed in the comment is based on an incorrect assumption in Moussallam et al. (2019). Since this point is related to Comment 2.7, we collectively state a response later under Comment 2.7.

Comment 2.6:

221. This temperature is equivalent to assume that only basalts at ridges produce volcanic gases, an unlikely scenario. What about volcanic gases emitted at subduction zones, which are colder by 200–300 °C? Or do you assume that subduction was not operating on a large scale prior to GOE? Please clarify.

Line 221: "temperature of 1473.15 K"

As the comment mentions, the degassing temperature at arc volcanism would be lower than we assumed, and such a low temperature lowers the K_{Oxy} and delays the oxic transition time (see Section S5.1 in the supplementary information). However, the complication in considering arc volcanism is that the island arc basalts are more oxidized than the mantle. So, the oxygenation effect of degassing via arc volcanism would result from the competing effects; i.e., more oxidized source melt and reduction due to low temperature.

To explain this point, we added a paragraph at line 285 in the main text and line 220 in the supplementary information.

Comment 2.7:

226. I don't know this paper but the available evidence tells rather the opposite (See Moussallam et al., 2019).

Line 226: “According to Kadoya et al. [57], higher mantle temperature in the past generally results in more oxidized volcanic gas”

Moussallam et al. (2019) present interesting data but unfortunately, the effect on the atmosphere implied by their data is misinterpreted because of a failure to consider redox conservation. As explained above (in our general comment for reviewers), reactions in a closed gas bubble do not change the net O₂ sink of the bubble owing to redox conservation while the reactions do change gas composition, f_{O_2} , and ΔFMQ of the bubble. Hence, the cooling due to ascent of the bubble fundamentally does not affect the results and conclusions about the Great Oxidation Event. Therefore, we can neglect this process to evaluate the K_{oxy} of the volcanic gas.

On the other hand, the cooling of the mantle reduces volcanic gas if the oxidation state, i.e., ΔFMQ , of the mantle is constant. This is simply because the f_{O_2} of the fayalite-magnetite-quartz buffer decreases with cooling (Fig. S.4). Hence, cooling also decreases f_{O_2} of the gas, resulting in more reducing volcanic gas.

We added paragraphs at line 254 and 263 to explain the observed difference in the oxidation state between erupted gas and its source and the reason in which we can neglect it. We also added a paragraph at line 275 to explain why the mantle cooling reduces volcanic gas. We also added a section to compare the results of Moussallam et al. (2019) and of this study in Section S.5.1.1.

Comment 2.8:

257. Perhaps say : The oxic transition time is older (larger)...

We moved and reworded the corresponding sentence in the supplementary information.

Comment 2.9:

360--361. This is what the vast majority believes which I also beg to disagree with. The fact that the redox state of magma source(s) is equal to that of the magma itself is an implicit yet debatable assumption (see Burgisser and Scaillet, 2007, *Nature*).

Aside from modification due to degassing (i.e., Moussallam et al., 2016), it is not clear that magmas are redox-fractionated from their source regions. It has been shown using both experimental data (Davis and Cottrell, 2018) and empirical studies of natural samples (Birner et al., 2018) that sources and residues of mantle melting are in redox equilibrium. The available evidence therefore shows that magmas do record the redox state of their source regions.

We added an explanation and the aforementioned citations at line 410.

Bruno Scaillet

Reviewer #3

The model presented here aims at evaluating quantitatively as to whether or not the oxidation of the mantle during the Hadean and the Archean may have caused the GOE. This is an important step forward that is appropriate for publication in Nature Communications. I have several point of criticisms that need to be addressed before the paper can be accepted for publication.

Comment 3.1:

The model relies on two main variables. The temporal change in the mantle f_{O_2} and the organic burial fraction (f_{org}) estimated from the mass balance of the carbon isotope record. Figure 2b shows that the inferred threshold of $K_{oxy}=1$ above which the atmosphere becomes permanently oxidized intercept the 95% confidence interval at about 2.5 Ga. However, this is critically dependent of the shape of the undulating curve, with an inflexion point located at $K_{oxy}\sim 0.85$ and ~ 2.65 Ga. Because this undulation is dependent on the temporal changes in f_{org} , this inflexion point should be linked to a change in f_{org} . As shown by Krissansen-Totton et al. (2015), however, there is no statistically significant difference between the pre-GOE and post-GOE f_{org} . Rather, there is a statistically significant increase in ϵ ($d^{13}C_{carb}-d^{13}C_{org}$), which is linked to the decrease in $d^{13}C_{org}$ in the late Archean, possibly due to methanotrophic-recycling. This is in contrast with the transition from the Proterozoic to the Phanerozoic, which is characterized by a well-constrained increase in f_{org} . How do these results fits with the curve shown in Figure 2b? Were both variables (f_{org} and ϵ) implemented and tested in the calculation of K_{oxy} ? How sensitive are these 2 variables on the location of the inflexion point at ~ 2.65 Ga?

In this study, we used the temporal change of f_{org} derived by Krissansen-Totton et al. (2015) as explained at line 181. Also, ϵ was used to calculate f_{org} . Note that $f_{org} = (\delta^{13}C_{carb} - \delta^{13}C_{in})/\epsilon$ (see Equation 1.3 in Krissansen-Totton et al. (2015)). Thus, not only the f_{org} but also ϵ has been implicitly considered in the calculation.

As the comment mentions, there is no statistical increase in f_{org} from average Archean to average Proterozoic (Krissansen-Totton et al., 2015). However, the f_{org} fluctuates in a shorter timescale of < 1 Gyr (Figure S.1), and such fluctuation affects the secular increase in K_{oxy} . The effect of fluctuation in f_{org} can be seen in the difference between Fig. 2a and 2b. Considering the fluctuation of f_{org} , the oxic transition time is 2.48 Ga (Fig. 2b) while setting the f_{org} at 20%, which is roughly the temporal average value of f_{org} , the oxic transition time is 2.62 Ga. Thus, relatively low value of f_{org} before 2.5 Ga (Fig. S.1) delays the oxic transition, but the sensitivity is relatively small compared to other processes discussed later and in Section S.5.

To highlight the effect of the fluctuation in f_{org} , we added an explanation at line 187.

Comment 3.2:

In the discussion section (lines 300--312), it is stressed that the uncertainty envelope allows an oxic regime before 2.4 Ga, which was not advocated in previous studies. Can be appropriate to refer to the study of Philippot et al. (2018, Nature Communications) who recently showed that the GOE initiated at least at ~ 2.45 Ga and most likely earlier.

Because of the simplicity of the model we used in this study, we didn't mean to suggest that the GOE occurred earlier than previously thought. Instead, the purpose of this study is to show the oxidation of the mantle (based quantitatively on new empirical data) can drive the

secular oxidation of the atmosphere. However, the manuscript was confusing, and it seemed as if we suggested the earlier rise of atmospheric oxygen.

So, we rewrote the discussion section to indicate that the oxic transition time depends also on other processes proposed by the previous studies. In addition, we cited Philippot et al. (2018) at line 215 to show the recent suggestion for the earlier GOE.

Comment 3.3:

The notion that the GOE may be the result of convection-induced homogenization of a redox-stratified mantle inherited from the disproportionation of Fe^{2+} into a Fe^{3+} rich phase stored in the lower mantle and metallic Fe^0 drained into the core has been proposed by Andraut et al., (2017, *Geochim. Persp. Let.*; missing reference to be added here). Their synchrotron-based experiments indicate a primordial Fe^{3+} excess of ~20% for the mantle iron, which would correspond to 500–1000 times the O_2 content in the present day Earth's atmosphere. How this estimate fits with the model presented here? Considering the quantitative approach proposed, it would be of major interest to develop on the total flux of free oxygen release to the atmosphere during the Archean-Proterozoic transition.

We cited Andraut et al. (2017) and added a discussion at line 300.

Much Fe^{3+} may be present but this oxygen never decouples from mantle mineral phases as molecular O_2 and so a direct comparison with the very small atmospheric reservoir (which is maintained by dynamic biological O_2 fluxes) could perhaps be described as a strained comparison, in our view. Data shows that volcanic gas contains negligible O_2 . Hence, the long-term source flux of O_2 is correctly represented in this paper as organic carbon and pyrite burial associated with photosynthesis and what matters is the competition of this O_2 source with efficient O_2 sinks represented by reduced volcanic gases (H_2 , CO , etc.). Thus, it's the effect of primordial Fe^{3+} on the source melt oxygen fugacity (f_{O_2}) and the way that the f_{O_2} changes the reduced and oxidized proportions of volcanic gases that is important. This concept is exactly the approach of our manuscript.

REVIEWERS' COMMENTS:

Reviewer #1 (Remarks to the Author):

The authors addressed all reviewers' comments and provided additional useful information about their calculations. I am not fully satisfied with considering volcanic gases and magma f_{O_2} as both reflecting the mantle source f_{O_2} . But, I must conclude that this is not the paper solving this enigma. Therefore, I strongly support the publication of this paper.

Best wishes,
Vincenzo Stagno

Reviewer #2 (Remarks to the Author):

The revised version reads well, as the first one. The paper is an interesting modelling exercise, which will be useful to scientists working on the oxygenation of the atmosphere, although not everyone will agree with model outcomes.

I have the following comments:

1. The authors now quote our work (Moussallam et al 2019) stating (line 258) that we claim that the cooling of the earth and ensuing oxidation of volcanic gases triggered the GOE. This formulation is incorrect. We said that such a change could have facilitated, not triggered, the transition toward oxidizing conditions along with other factors, because we are quite aware that this important transition was possibly driven by a number of mechanisms. The authors state the same in their paper in press in GPL but I guess that it is too late at this stage to correct this overstatement.
2. The authors claim that we did a conceptual error in our analysis. I obviously do not agree. What we did is simply to use the f factor derived by Holland to illustrate how the change in gas speciation as affected by temperature changes this parameter (which could be wrong, but this is another issue). The gas data we show are what they are, as long as one believes they represent some equilibrium. It is, in my opinion, a matter of what temperature you select to mix your volcanic gases with the ambient atmosphere. This is just a comment however, since that part belongs more to the GPL paper.
3. On line 129, it is said that basalts are more evolved than komatiite and this would result in these melts being more reduced than komatiite due to sulfur degassing. Please be more specific? Is it because you believe that basalts are richer in S than komatiite?
4. lines 276-278. As I understand it, your melt buffered stage assumes that nothing happens (in terms of redox evolution) during decompression. Obviously, I also disagree here, in view of the growing evidence provided by several groups in various settings on that matter (see last paper of Moussallam et al on hot spots). This is the main weakness, and quite a severe one, of this otherwise interesting paper.
5. line 313. I guess you want to say decomposition of water? (not the composition)

Bruno Scaillet.

Reviewer #3 (Remarks to the Author):

I am satisfied with the changes made in this revised manuscript.

We thank the reviewers for continuing support to improve our manuscript and for volunteering their time. Here, we have modified our manuscript following the comments with changes that are highlighted in **blue color**. In addition, following the editor's instruction, we combined the method sections that had been in the main text and the supplementary information. Below reviewer comments are in normal typeface, and our responses and comments are in **bold blue typeface**.

REVIEWERS' COMMENTS:

Reviewer #1 (Remarks to the Author):

The authors addressed all reviewers' comments and provided additional useful information about their calculations. I am not fully satisfied with considering volcanic gases and magma fo2 as both reflecting the mantle source fo2. But, I must conclude that this is not the paper solving this enigma. Therefore, I strongly support the publication of this paper.

Best wishes,
Vincenzo Stagno

Reviewer #2 (Remarks to the Author):

The revised version reads well, as the first one. The paper is an interesting modelling exercise, which will be useful to scientists working on the oxygenation of the atmosphere, although not everyone will agree with model outcomes.

I have the following comments:

1. The authors now quote our work (Moussallam et al 2019) stating (line 258) that we claim that the cooling of the earth and ensuing oxidation of volcanic gases triggered the GOE. This formulation is incorrect. We said that such a change could have facilitated, not triggered, the transition toward oxidizing conditions along with other factors, because we are quite aware that this important transition was possibly driven by a number of mechanisms. The authors state the same in their paper in press in GPL but I guess that it is too late at this stage to correct this overstatement.

We are sorry about misquoting your paper on this important subject.

We modified the statement you refer to at line 264 as follows:

“As such, it was proposed that secular cooling of the mantle could have facilitated the GOE”

2. The authors claim that we did a conceptual error in our analysis. I obviously do not agree. What we did is simply to use the f factor derived by Holland to illustrate how the change in gas speciation as affected by temperature changes this parameter (which could be wrong, but this is another issue). The gas data we show are what they are, as long as one believes they represent some equilibrium. It is, in my opinion, a matter of what temperature you select to mix your volcanic gases with the ambient atmosphere. This is just a comment however, since that part belongs more to the GPL paper.

The point we are skeptical about is that the redox balance seems not to be satisfied in Moussallam et al. (2016) though they assumed a closed system. As we discussed in Kadoya et al. (2020), while cooling changes the gas speciation, resulting in a change in K_{oxy} (and Holland's f factor), it cannot change the redox balance, in particular, the O_2 sink of the gas mixture. Hence, the overall oxygenation effect of the closed gas mixture cannot change, i.e., K_{oxy} (and Holland's f factor) cannot flip from >1 to <1 or vice versa.

In addition, the temperature under which a volcanic gas mixture mixes with the ambient atmosphere is not the key issue. This is because mixing of volcanic gas with atmosphere means that the volcanic gas affects the atmosphere. To evaluate the oxygenation effect of volcanic gas on the atmosphere, it is necessary to investigate the state of the volcanic gas before it mixes with the atmosphere.

We added Supplementary Note 2 to explain that the Holland's f factor has the same temperature dependence with K_{oxy} .

3. On line 129, it is said that basalts are more evolved than komatiite and this would result in these melts being more reduced than komatiite due to sulfur degassing. Please be more specific? Is it because you believe that basalts are richer in S than komatiite?

Thank you for bringing up this important issue. Komatiites are high-temperature, high degree partial melts that are inherently undersaturated in sulfur. This conclusion is applicable to all the komatiite and most picrite systems in Nicklas et al. (2018, 2019), and this issue has been specifically addressed in those publications. The most convincing evidence for the sulfur-undersaturated nature of the komatiitic and picritic lavas is the fact that the strongly chalcophile elements, such as Cu, Ni, and Pd, behaved incompatibly during differentiation of these lavas upon emplacement. This can be seen in the MgO vs. Cu, Ni, and Pb diagrams, where the data plot on olivine control lines, indicating that olivine was the only fractionating phase over the entire compositional range of the lavas. Upon sulfur saturation, these elements are tied up in immiscible sulfide liquid that fractionates in the komatiitic/picritic melt; as a result, the data deviate dramatically from olivine control. Hence, komatiitic lavas cannot outgas sulfur upon emplacement, and their redox cannot be affected by this process.

Basalts, on the other hand, are much lower degree melts that are evolved and, by the same token, sulfur-saturated melts. When erupted subaerially, basalts and some picrites have been known to experience sulfur outgassing, which potentially could have modified their redox state.

To clarify this point, we modified the sentence at line 130.

4. lines 276-278. As I understand it, your melt buffered stage assumes that nothing happens (in terms of redox evolution) during decompression. Obviously, I also disagree here, in view of the growing evidence provided by several groups in various settings on that matter (see last paper of Moussallam et al on hot spots). This is the main weakness, and quite a severe one, of this otherwise interesting paper.

We don't exclude the possibility that during the melt buffer stage, degassing affects both of redox states of gas mixture and its surrounding melt. However, the key point of Kadoya et al. (2020, *GPL*) is that the atmospheric *oxygenation effect* of a volcanic gas mixture

is determined by the oxygen fugacity at the time which the gas mixture decouples with its surrounding melt and that the change during the melt buffer stage does not affect it.

Before the decoupling, i.e., during the melt buffer stage, the gas mixture interacts with the surrounding melt by definition. Hence, the oxidation of the gas mixture should accompany reduction of the melt as data suggest. However, the oxygen fugacity of the gas mixture and melt would be equal because they interact with each other. So, decompression should decrease the temperature and may change the speciation and f_{O_2} of the gas mixture. But the f_{O_2} of the gas mixture would equal to that of the surrounding melt as long as the gas and melt interacts. Therefore, the oxygenation effect of the gas mixture can be estimated using the f_{O_2} of the surrounding melt.

On the other hand, after the gas mixture decouples with the surrounding melt, i.e., during the closed stage, the decompression that decreases temperature changes the f_{O_2} of the volcanic gas independently of the change in the melt. In particular, relative to the FMQ buffer, the cooling tends to increase ΔFMQ of the volcanic gas mixture as both Moussallam et al. (2016) and Kadoya et al. (2020) mentioned. However, the important point is that any reaction during the closed stage does not change the oxygenation effect of the gas mixture because of the redox balance as explained in Kadoya et al. (2020).

Therefore, the oxygenation effect of a volcanic gas can be estimated using the f_{O_2} at which the gas decouples with its surrounding melt.

It is also noteworthy that both Aulbach et al. (2016) and Nicklas et al. (2019) indicate a similar trend of f_{O_2} evolution even though they measured different type of rocks. This similarity indicates that the f_{O_2} of the melt has the same trend despite of the difference in the degree of partial melting. It supports our assumption that the f_{O_2} of the volcanic gas at decoupling from melt secularly increases with time.

The remaining caveat is the absolute value of the f_{O_2} of the gas at the decoupling. This point is partly addressed by the sensitivity study of the anchoring value of the modern f_{O_2} of the mantle. To highlight this point, we add paragraphs at line 307.

5. line 313. I guess you want to say decomposition of water? (not the composition)

Yes, thank you. We modified the wording accordingly.

Bruno Scaillet.

Reviewer #3 (Remarks to the Author):

I am satisfied with the changes made in this revised manuscript.